# Investigation of the Effect of Vegetation on Flow Structures and Turbulence Anisotropy around Semi-Elliptical Abutment

**Seyedeh Fatemeh Nabaei** [1], **Hossein Afzalimehr** [2,*], **Jueyi Sui** [3,*], **Bimlesh Kumar** [4]
**and Seyed Hamidreza Nabaei** [5]

1   Department of Water Engineering, Isfahan University of Technology, Isfahan 8415683111, Iran; f.nabaei@alumni.iut.ac.ir
2   Department of Civil Engineering, Iran University of Science and Technology, Tehran 1684613114, Iran
3   School of Engineering, University of Northern British Columbia, Prince George, BC V2N 4Z9, Canada
4   Department of Civil Engineering, Indian Institute of Technology Guwahati, Guwahati 781039, India; bimk@iitg.ac.in
5   Department of Mechanical Engineering, Tabriz University, Tabriz 5166616471, Iran; hamidreza.nabaei95@ms.tabrizu.ac.ir
*   Correspondence: hafzali@iust.ac.ir (H.A.); jueyi.sui@unbc.ca (J.S.); Tel.: +1-250-960-6399 (J.S.)

**Abstract:** In the present experimental study, the effect of vegetation on flow structure and scour profile around a bridge abutment has been investigated. The vegetation in the channel bed significantly impacted the turbulent statistics and turbulence anisotropy. Interestingly, compared to the channel without vegetation, the presence of vegetation in the channel bed dramatically reduced the primary vortex, but less impacts the wake vortex. Moreover, the tangential and radial velocities decreased with the vegetation in the channel bed, while the vertical velocity (azimuthal angle > 90°) had large positive values near the scour hole bed. Results showed that the presence of the vegetation in the channel bed caused a noticeable decrease in the Reynolds shear stress. Analysis of the Reynolds stress anisotropy indicated that the flow had more tendency to be isotropic for the vegetated bed. Results have shown that the anisotropy profile changes from pancake-shaped to cigar-shaped in the un-vegetated channel. In contrast, it had the opposite reaction for the vegetated bed.

**Keywords:** local scour; semi-elliptical abutment; vegetated bed; Reynolds shear stress; turbulence intensity; Reynolds stress anisotropy

## 1. Introduction

Over the past several decades, researchers have emphasized the importance of the scour process for designing bridges and other hydraulic structures, e.g., [1–5]. One of the crucial causes of bridge failure is the scour around its abutment and the associated flow pattern around the structure [6]. The Federal Highway Administration of the USA investigated 383 cases of bridges damaged by catastrophic floods and estimated that 72% of the bridge failures were caused by abutment damage [7]. The flow pattern and scour mechanism around the bridge abutment are phenomena that result from the reaction between the three-dimensional turbulence flow field around the abutment foundation in erodible beds [8]. Around the abutments, the development of the boundary layer of the protrusion wall has created complexity in the flow field [9]. The flow field around an abutment involves a complex three-dimensional (3D) vortex flow, and this complexity is increased by the development of the scour hole, involving flow separation [10]. The scour hole around an abutment is developed by both primary vortices and the downflow, similar to a horseshoe vortex at piers. The downflow is the principal cause of the development of the scour hole. The secondary vortexes are created near the primary vortex, behind the abutment and at the separation zone, by limiting the power of the primary vortex in the scour hole development. Downstream of the abutment, the factor that causes the

flow separation from the abutment creates the wake vortices [11]. In short, it can be said that the impact of the flow on the upstream face of the abutment and the separation of it downstream of the abutment is one of the most critical factors in the scouring process at the abutments [10]. By studying wing-wall abutment, Kw An and Melville [11] found that the wake vortices downstream of an abutment were caused by the flow separation at the abutment's corner. As such, the wake vortices form at the downstream region of the abutment. These vortices, with the vertical axis and low-pressure center, suck up sediment particles and move sediment particles downstream, after separation from the bed with the mainstream, creating an independent scour hole downstream of an abutment [10]. Readers can refer to figure 6.3 of Melville and Coleman [12], which illustrates the flow and scour patterns around a short abutment.

There have been numerous investigations for estimating the local scour rate around bridge abutments, e.g., [13–16]. Some researchers have also studied the flow field and characteristics of flow around bridge abutments within scour holes [3,17,18]. Most of these studies are concentrated on flow patterns around bridge abutments in an alluvial channel based on laboratory experiments.

Due to the complexity of the scouring process around an abutment, resolving the flow feature near the scour hole bed and turbulence characteristics is profoundly challenging [19]. The assumption of isotropic distribution of turbulent statistics in numerical models prohibits their application in scouring around the bed abutment. The purpose of conducting anisotropy analysis is to understand the turbulent flow characteristics better and determine the turbulence structure's sensitivity for different bed conditions [20–22]. By introducing the invariant functions, the turbulence anisotropy effectively reduces the complexity of a three-dimensional flow field to a two-dimensional flow that is simpler for analysis [23]. Thus, the Reynolds stress anisotropy study is an important research topic for developing turbulence theories and numerical simulations.

Lumley and Newman [20] proposed the technique of the anisotropy invariants, which provides a procedure to analyze turbulence. Their study used the normalized Reynolds stress anisotropy tensor invariants to establish the anisotropy invariant map (AIM, also referred to as the Lumley map). Mera et al. [24] analyzed the evolution of the magnitude and nature of anisotropy along a meandering river. AIM also characterized the spatial distribution of anisotropy invariants and the nature of turbulence anisotropy.

The riverine plants play a crucial role in river dynamics and hydraulic structure [25]. From a hydrological view, although it is of utmost importance to protect the ecosystem area of a river network dominated by vegetation [26], this issue has not been thoroughly studied and used effectively. Besides, some engineering measures and expensive solutions have been installed to prevent a local scour process around hydraulic structures (such as the collar, spur dike, and submerged vane). On the other hand, the application of vegetation for reducing scouring can provide an eco-friendly and economical outcome, as it can help with the protection of the river territory, and it is more compatible with the environment. Luhar et al. [27] used hydrodynamic concepts to explore how vegetation affects the flow and transmission and how the flow feedback can affect the spatial structure of vegetation. Some studies have been conducted to describe the flow in vegetated channels. Results indicated that the presence of vegetation in the channel bed modifies flow and sediment transport [28–30]. In these studies, it was found that vegetation in the channel increases flow resistance, modifies flow patterns, and provides additional drag while leading to a decrease in the bed shear stress, thus significantly decreasing sediment transport.

Afzalimehr et al. [31] stated that in channels with a submerged vegetated bed, its Reynolds stress distribution was nonlinear due to the presence of drag force. This drag force was caused by submerged plants in the water and had its maximum value slightly above the top of the vegetation canopy. Comparing Reynolds stress distribution and turbulence intensity graphs to those of velocity distribution in non-uniform decelerating flow with a vegetation bed, Keshavarz et al. [32] concluded that the zone for the maximum Reynolds stresses and turbulence intensity coincides with the region where the highest rate of change

in velocity occurs. Afzalimehr et al. [33] asserted that vegetation on channel banks changes the shape of the scour hole compared to that without vegetation. They concluded that vegetated banks decreased Reynolds shear stress near the bed and played no role in the vortex structure downstream of the abutment. In such circumstances, understanding the effect of vegetation on flow conditions in river restoration projects is essential. Although some studies have been reported, knowledge gaps regarding the impact of vegetation around the hydraulic structure, such as the bridge abutment, remain.

The vegetation in a channel bed is one of the prominent factors in environmental hydraulics, which impacts the flow field and scouring process around the abutments. In this experimental study, a semi-elliptical abutment has been used because of its aerodynamical feature. The purpose of this study was to investigate the flow pattern around a semi-elliptical abutment in a channel with submerged vegetation (hereafter, we use the word vegetation to represent the submerged vegetation) and the impact of vegetation on the flow pattern. The effect of the anisotropic turbulence on the local scour has been introduced in the present work. In the present study, equivalent experimental conditions have been set to compare the effect of a vegetated channel bed on the local scour around an abutment to that without a vegetation cover. The following aspects have been investigated: the scouring mechanism, changes of the flow parameters, flow field, Reynolds shear stress distribution, Reynolds stress anisotropy, and evolution of the anisotropic invariant function around the semi-elliptical abutment.

## 2. Materials and Methods

Experiments have been conducted in a laboratory flume for two cases: channel bed without vegetation (first case) and channel bed with vegetation (second case). The flume is 16.0 m long, 0.9 m wide, and 0.6 m deep with a rectangular cross-sectional area. Experiments for each case have been repeated twice under the same hydraulic conditions. The flume has a glass floor with glass walls. The bed slope of this flume is 0.0003. A weir located at the end of the flume was used to adjust the water level in the flume. A pump was used to circulate the water with a maximum discharge capacity of 0.06 $m^3$ $s^{-1}$ from the sump. An electromagnetic flowmeter was installed in the supply conduit to measure the discharge passing through the flume continuously. A semi-elliptical abutment has been used in this experimental study. The semi-elliptical abutment has a length (along the streamwise direction) of $l$ = 15 cm and a width (perpendicular to the streamwise direction) of $b$ = 4 cm.

The abutment was made of Teflon (thermoplastic polymer), which has suitable properties such as lightness, high strength, and long-term durability in water. The selection of the abutment ($b$) width should be sufficient to avoid the influence of the channel banks on the scouring process around it (this means the sidewall has no effect on the scour in this case study). Chiew and Melville [34] stated that the distance ratio of the facing wall from the center of the abutment to the width of the abutment ($b$) is equal to at least 5. A semi-elliptical abutment was chosen in this study, since a few studies have been performed by using a semi-elliptical abutment [3,33]. The experimental section (sandbox) was 1.0 m long, 0.16 m deep, and 0.9 m wide, located at 10.3 m downstream of the flume entrance. In this testing section, the flow was developed. The abutment was embedded in this sandbox. Teflon panels were installed at 0.16 m above the flume bottom and covered with sediments. The one-meter-long sandbox between the upstream Teflon panel and the downstream Teflon panel was filled with sieved sand. According to Dey et al. [35], to eliminate the non-uniform effect of sediment particles that can significantly reduce the scour depth, the geometric standard deviation of the sediment particle size $\sigma_g = (d_{84}/d_{16})^{0.5}$ should be less than 1.4. In this equation, $d_i$ is the grain size (mm), smaller than $i$ percent of sediment particles. The sediment used in this study was uniform sand prepared by using mechanical sieve analysis tests. The median diameter of the sediment particles was $d_{50}$ = 0.75 mm, and the geometric standard deviation of the particle size distribution $\sigma_g$ was 1.22 (<1.4).

For this experimental study, the discharge was set up as $Q = 0.05$ m$^3$ s$^{-1}$ with a flow depth of 0.23 m in the flume, at a point where no movement of sediment was observed in the channel bed. As mentioned before, the flow depth in the flume was adjusted by the end slide gate. Moreover, under the condition of the same flow rate, water depth was gradually decreased, and velocity gradually increased until the motion of sediment particles was observed, namely, the incipient motion of sediment particles. The subcritical flow depth of 0.18 m for the incipient motion of sediment particles was obtained. Then, the approaching flow depth $h$ was maintained at 19 cm. The experiment ran with an average approaching flow velocity of U = 0.292 m s$^{-1}$ and Froude number of Fr = 0.2138 for each experiment, which satisfied the clear water scour condition of $u^*/u^*_c = 0.95$, where $u^*$ is the shear velocity, which is calculated using the log law [36,37]. In this experimental study, $u^*_c$ and $u^*$ values were 0.0126 m s$^{-1}$ and 0.01197 m s$^{-1}$, respectively. The point gauge with an accuracy of $\pm 1.0$ mm was used to measure flow depth.

In the first phase of this experimental study, the abutment was installed at 11.0 m downstream from the flume entrance. The flow in the testing section (sandbox) was fully developed turbulent flow, since it was confirmed by the longitudinal velocity profiles measured at the cross-sections of 7.5, 8, and 8.5 m from the flume entrance. At these three cross-sections upstream of the sandbox, the longitudinal velocity profiles had been matched to each other.

In the second phase of this experimental study, the effect of vegetation in channel bed on the flow characteristics around the semi-elliptical abutment was investigated. To the best of our knowledge, neither experimental nor numerical studies have been carried out in this field. In the first step, a glass panel with dimensions of 0.4 m × 0.5 m × 0.006 m (length × width × thickness) was placed on the bottom of the sandbox. Then, numerous bunches of vegetation were stuck on the glass bed. These vegetation are nearly uniformly distributed around the abutment and form a geometrical network in the form of rhombuses with equal dimensions and a side length of 14 cm (approximately) (Figure 1). Each vegetation bunch consists of about 55–65 pine needles.

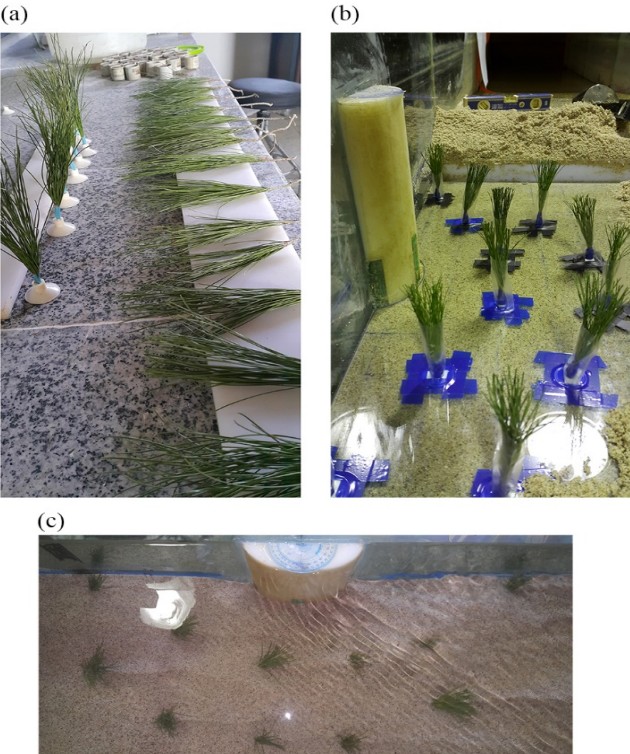

**Figure 1.** (**a**) Vegetation bunches consisting of pine needles used in experiments; (**b,c**) vegetation bunches installation and arrangement.

In the present study, this type of geometrical distribution of vegetation has been selected to consider the effect of resistance created by the scattered vegetation spots in the flow and understand how it affects the scouring process and flow structure. Pine needles were used as vegetation because of their resistance, stability, and woodiness [38]. The height of the pine needles was 0.15 m. After installing the vegetation on the glass panel, the vegetation bed was filled with sieved sand with a depth of 0.11 m. The length of the vegetation above the sediment bed was only 0.04 m. With such a setup of vegetated bed around the semi-elliptical abutment, the experiment was run under clear water scour conditions with defined discharge.

According to Melville and Chiew [39], the scour equilibrium was achieved after 72 hours in the first experiment for channel bed without vegetation and 48 hours in the second experiment for channel bed with vegetation (Figure 2).

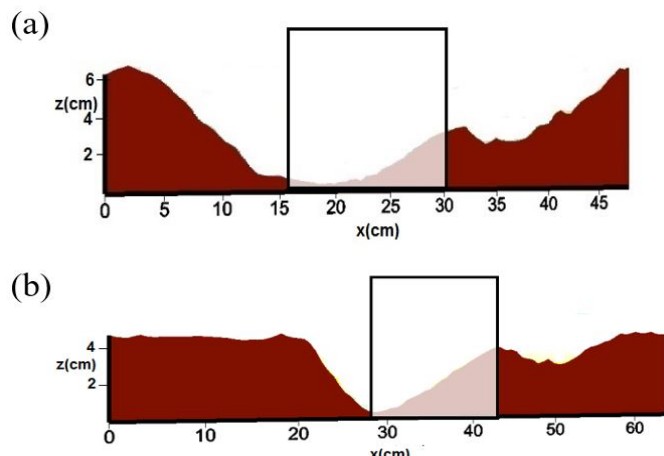

**Figure 2.** Profile of the scour hole after the equilibrium state was achieved (**a**) without vegetation and (**b**) with vegetation.

After the scour process reached the equilibrium state for each experimental run, the flow discharge was slowly reduced to zero, and the pump was switched off. After the flume water was completely drained, the scour hole was stabilized by spraying a thin layer of chemical material. A hypothetical network with a resolution of 1.0 cm was determined on the bed surface, and their intersection points were marked. Then, about 1500 points were measured using a mobile limnimeter (a point gauge). After that, by defining an origin mark (shown in Figure 3, with the coordinate (0,0,0)), the coordinates for the collected data were applied to each point, and the topography was drawn using Surfer software.

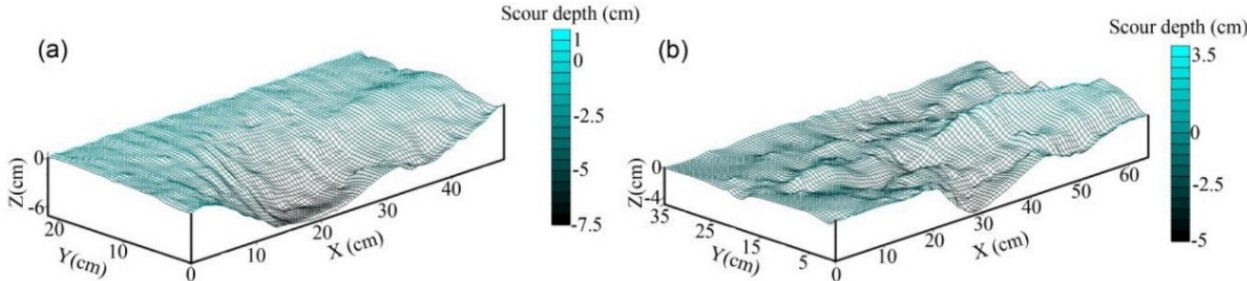

**Figure 3.** Topography of the final scouring hole around the abutment: (**a**) without vegetation (abutment coordinates: x = [16,31], y = [0,r], z= [0,40]); (**b**) with vegetation (abutment coordinates: x = [28,43], y = [0,r], z = [0,40]).

Figure 3 shows the topography of the flume bed after the scouring process reaches the equilibrium state. After measuring the scoured flume bed, the flume was gradually filled up with water again. Then, velocities at different azimuthal angles $\theta$ with specific radial distances $r$ were measured. The instantaneous three-dimensional velocity components were measured at different sections using a down-looking acoustic Doppler velocimeter (ADV), Vectrino$^+$ model, made by Nortek, with a duration of 120 s. The sampling frequency was set at 200 Hz [40]. The accuracy and quality of the collected data were controlled by two parameters, the correlation coefficient (COR) and the signal-to-noise ratio (SNR).

In this study, to investigate the flow patterns around the abutment, it was necessary to measure the velocities around the scour holes at different azimuthal sections. Therefore, we selected the measured points in cylindrical polar coordinates ($\theta$, $r$, $z$) with azimuthal angles of 30°, 60°, 90°, 120°, and 160°. Additionally, using the ADV at each azimuthal angle, four profiles were measured at the radial axis with spacing distances of 4, 6, 8, and 10 cm from the abutment's surface. Along each vertical axis, data have been collected at 20–30 measuring points. Observations have been collected from 4 mm above the bed to the point 5 cm below the water surface.

Win ADV software [41] has been used to calculate turbulent flow statistics. The software also filters the inappropriate recorded data with SNR and COR less than 15 dB and 70%, respectively. In this study, we used this filter to obtain the desired data. Moreover, the filter provided by Goring and Nikora [40] for the phase-space threshold despiking has been used to detect and eliminate the spurious data.

## 3. Results and Discussions

The scouring process was initiated from the front face of the abutment in an unvegetated channel. The two-dimensional stream acceleration primarily causes scouring due to the effect of obstacles. Then, rill erosion was quickly formed around the abutment [42,43]. A few minutes later, a creeping motion of bed materials happened. The rill erosion reached the rear of the abutment, and the scouring process extended to the downstream region of the abutment. In other words, the shear stress in the vicinity of the abutment exceeds its critical value, hence causing bed erosion around the abutment. It was observed that sediment particles were transported further downstream of the abutment and progressively formed a deposition dune. This phenomenon can be explained as the washing out process of the bed particles in the front face of the abutment by the downward flow, which causes the development of the scour hole around the abutment. Besides, the presence of the wake vortices influences the movement of bed particles, which results in decreasing the dune's height. After the initial scouring process, the primary vortex (horseshoe vortex) structures seem to play a significant role in extending and developing the scour hole around the abutment. This process was observed in the case of the vegetated bed as well, but with less equilibrium scour depth. The maximum scour depth for the case without vegetation in the bed was located near the abutment nose (in the middle portion and upstream of the abutment) and reached 6.6 cm.

However, for the case with a vegetated bed, the maximum scour depth moved toward the front face of the abutment and reached a maximum depth of 4.3 cm. Results showed that the appearance of vegetation on the channel bed resulted in a noticeable reduction in the depth of the scour hole. Moreover, since the vegetation have significantly changed the flow velocity, the incipient motion process of sediment has been affected. The results of the incipient motion process around the abutment in a vegetated channel confirmed the findings of Shahmohammadi et al. [44,45], who studied the incipient motion of sediment in a vegetated channel without the presence of an abutment. Shahmohammadi et al. [44,45] claimed that the presence of vegetation patches resulted in a decrease in cross-averaged streamwise velocity by almost 20%. Moreover, the scouring process around multiple vegetation patches interacted with and affected the transport of particles from the vegetation patch to further downstream of the channel. In the present study, the eroded sediment

particles were delivered downstream of the abutment to form the highest deposition dune at a 15 cm distance from the channel wall.

The instantaneous changes of the scour depth around the abutment for both the vegetated and un-vegetated bed are shown in Figure 4. One can see from Figure 4 that the presence of bed vegetation significantly reduced the depth of the scour hole. Up to $t$ = 120 min into the scour process (the intersection of two trend lines for scour depth), the scour depth for the case with the vegetated bed was surprisingly higher than that of the un-vegetated bed. During the initial stage of the scouring process, in the vicinity of the upstream region of the abutment, the vegetation resulted in stronger primary vortices. After 2 h of the experimental run, the scour depth for the case without vegetation in the bed increased faster than for the case with the vegetated bed. For both cases, the scour hole depths increased gradually with time but at a decreasing rate.

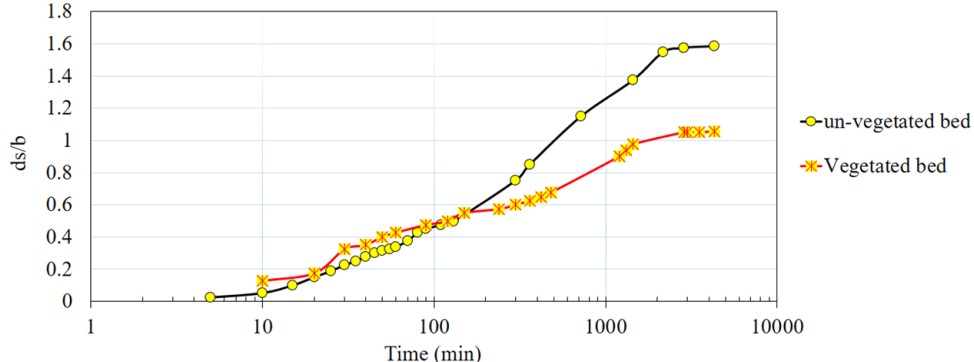

**Figure 4.** Variation of relative scour depth with time in the front face of the semi-elliptical abutment for both un-vegetated and vegetated beds.

### 3.1. Velocity Field around the Abutment

With the increase in the scour depth, the flow velocity decreased, since the flow cross-sectional area increased. Figure 5a,b show the streamlines for the un-vegetated bed and Figure 5c,d for the vegetated bed, after the scouring process reached the equilibrium state. The streamlines were drawn using a linear interpolation method at two heights of $d$ = 0.6 cm and $d$ = 10.0 cm. It was also observed that water moved slower with less momentum near the abutment wall and its base and then turned away more than the faster-moving water from the abutment and the bed. This phenomenon caused the skewing of velocity profiles, as explained by Dey [10]. Upstream of the abutment, close to the scour hole bed ( mboxemph$d$ = 0.6 cm), the circulation was strong and decreased with an increase in the azimuthal angles $\theta$. The horizontal over the scour hole resulted in a reverse flow and wakes in the scour hole. Figure 5 shows that the appearance of vegetation around the abutment influenced the mean velocity field ($U$) patterns in both heights ($d$ = 0.6 cm and $d$ = 10.0 cm from the scour hole bed, respectively). It was observed that vegetation significantly reduces the mean velocity around the abutment at a lower height ($d$ = 0.6 cm). This phenomenon was observed in the downstream area of the abutment (Figure 5c). Close to the scour hole bed ($d$ = 0.6 cm), a significant decrease in the mean velocity was observed at the point ($x, y$) = (4 cm, 9 cm) for the case without vegetation in the bed and at ($x, y$) = (1 cm, 14 cm) for the case with the vegetated bed. The $U$ values reached their minimum at the scour hole bed and increased with the distance from the scour hole bed.

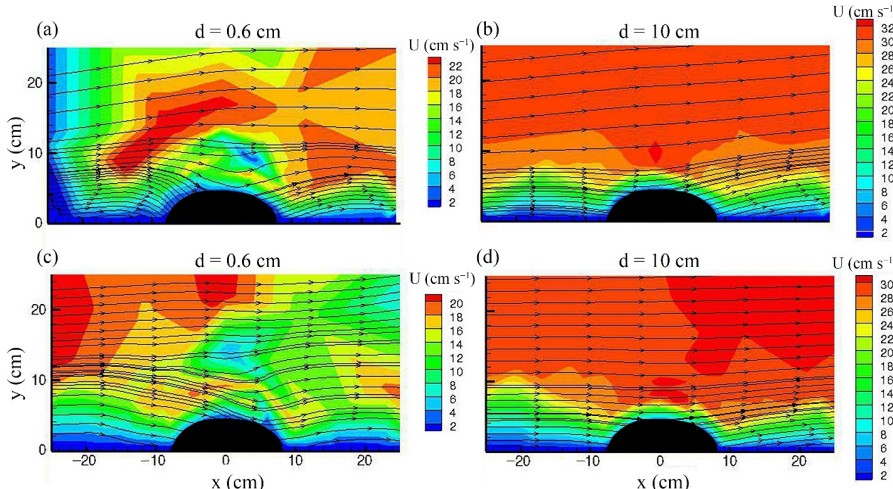

**Figure 5.** Streamlines around the abutment (**a,b**) for the un-vegetated bed and (**c,d**) for the vegetated bed at two heights of $d$ = 0.6 cm and $d$ = 10.0 cm from the stabilized scour bed. Note that the range of the color scale is not equalized for better illustration.

A cylindrical polar coordinate (Figures 6–11) was used to represent the flow patterns. The time-averaged velocity components in ($\theta$, $r$, $z$) are respectively represented by ($u$, $v$, $w$), whose corresponding fluctuations are ($u'$, $v'$, $w'$). For this experimental study, $u$ describes the tangential velocity, $v$ is used to denote the radial velocity, and the vertical velocity is expressed as $w$. The positive directions of $u$, $v$, and $w$ are counterclockwise, outward, and upward, respectively. The velocity distributions are plotted in the $rz$ plane at different azimuthal angles $\theta$ of 30°, 60°, 90°, 120°, and 160°. The abutment wall represents $r_0$ = 0, and $r_0$ refers to Equation (1) for calculating the abscissa scale.

$$r_0 = r - r(\theta) = r - \frac{bl/2}{\sqrt{(b\cos\theta)^2 + (l/2\sin\theta)^2}} \tag{1}$$

For the case without vegetation in the bed, the contours of the tangential velocity $u$ at different azimuthal planes (30°, 60°, 90°, 120°, and 160°) are shown in Figure 6. One can observe the features of the passage of the flow around the abutment. The tangential velocity $u$, among all 3D velocity components, played a crucial role in developing the scour hole and the turbulence structure. The magnitude of $u$ without vegetation increased with the azimuthal degree $\theta$ from 0° to 120° and then decreased at $\theta$ = 160° (at the downstream zone of the abutment). The magnitude of $u$ was more significant when the scour depth was smaller, while it decreased gradually ($\theta$ = 160°) with an increase in scour depth because of an increase in flow area. The strong circulation in the upstream region of the abutment caused a significant decrease in the tangential velocity to a great extent near the scour bed. The results clearly showed that this impact had been reduced with the distance from the abutment. In Figure 6, negative values have not been observed. However, by screening data obtained using the ADV, the negative instantaneous tangential velocities have been noticed. Either vortex or turbulence may cause these negative instantaneous tangential velocities. The magnitude of $u$ increased in the vertical direction from the scour bed, with the exception of near the water surface for 160°. The results showed that the velocity gradient ($\partial u/\partial z$) inside the scour hole ($z < 0$) was more than that of the outside of the scour hole ($z \geq 0$). In the region downstream of the abutment, due to the effect of the wake vortex, $u$ was declined near the water surface at $\theta$ = 160°. The contour lines of tangential velocity were concentrated near the scoured bed, especially at the edge of the scour hole, indicating the rapid change of $u$, namely a high-velocity gradient ($\partial u/\partial z$).

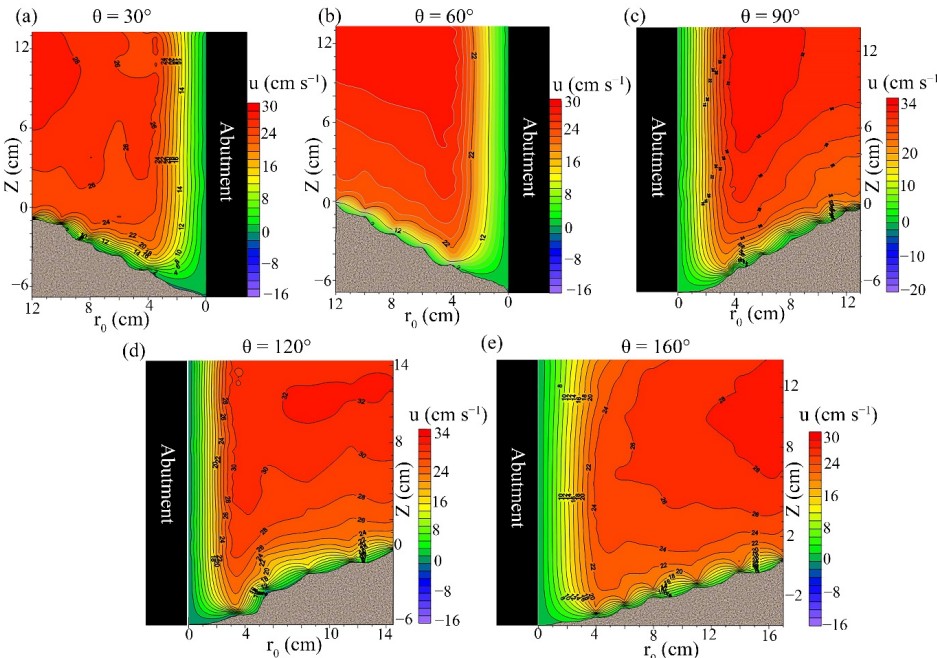

**Figure 6.** Velocity field for the case bed without vegetation in the tangential direction (*u*, cm/s) for different azimuthal sections. Note that the range of the color scale is not equalized for better illustratio.

The scour depth decreased significantly from 6.6 cm in the middle of the abutment to 4.3 cm in the front of the abutment, owing to less tangential velocity (Figure 7). In other words, vegetation cover in the channel bed resulted in different velocity patterns that generated different scouring processes. The tangential velocity decreased with the increase in the radial distances outside the scour hole, compared to the un-vegetated channel. This finding is similar to the observation of Yamasaki et al. [46]. The decrease in the streamwise velocity resulted from the adjacent vegetation's effect; a small deposition ridge was formed along the centerline. By comparing Figure 6 to Figure 7, regardless of whether there were vegetation bunchlets in the channel bed, the tangential velocity was positive. This finding confirmed the result reported by Jafari and Sui [47]. For the channel bed without vegetation, the tangential velocity component decreased rapidly near the scour hole bed, especially at the upstream face of the abutment. With the presence of vegetation, a region has been observed at the upstream face of the abutment. The velocity contour lines first concentrated with lower tangential velocity and then moved away from each other (Figure 7). This deceleration spot and its changes were firstly observed outside the scour hole ($\theta = 30°$) and then, with the increase in azimuthal angle, it moved inside the scour hole ($\theta = 60°$, $90°$) and approached the abutment wall. Although the tangential velocity increased with the azimuthal angle, the deceleration rate at $\theta = 90°$ was more significant in the marked spot (black circle) than those at $\theta = 30°$, $60°$.

Consequently, the maximum scour depth moved from the abutment nose to the abutment tip (corner of the abutment attached to the wall). In other words, the presence of vegetation caused part of the deceleration to occur above the original bed level. Thus, the scouring depth was reduced.

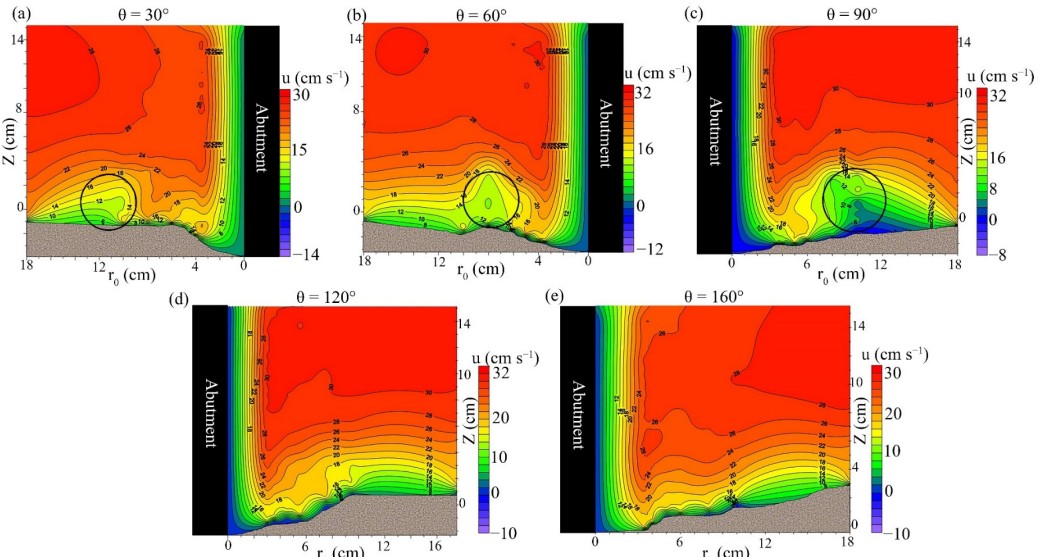

**Figure 7.** Velocity field for the case with a vegetated bed in the tangential direction ($u$, cm/s) at different azimuthal sections. Note that the range of the color scale is not equalized for better illustration.

Figure 8 shows a comparison between tangential velocity profiles for the vegetated bed and un-vegetated bed at azimuthal planes (30°, 60°, 90°, 120°, and 160°) at different distances from the abutment surface (4, 6, 8, and 10 cm). From Figure 8, it can be seen:

(1) The larger the distance from the abutment in the radial direction, the more the skewness in the distribution pattern of tangential velocity, especially at the upstream face of the abutment in an un-vegetated channel. Additionally, with the increase in the azimuthal angle (moving downstream), the skewness decreased. In other words, by moving downstream, the tangential velocity near the scoured bed tended to increase.

(2) For the vegetated bed case, the tangential velocity profile distribution had an "S" shape (changed near the scour bed), and this can be seen at all azimuthal angles, except the 160°.

(3) With the presence of vegetation in the channel bed, the tangential velocity profile has two turning points caused by the interplay between the drag force, the gravitational potential, and the momentum flux gradient. Such a result has been reported by Huai et al. [48] and Shahmohammadi et al. [44].

(4) For the case with a vegetated bed, the velocity gradient near the channel bed became negative with the increase in the radial distance from the abutment, especially between 5 cm and 12 cm at all azimuthal angles, except the 160°. With the increase in the azimuth angle, the tangential velocity increased slower than the case without vegetation in bed (Mostly at $z < 0.3h$).

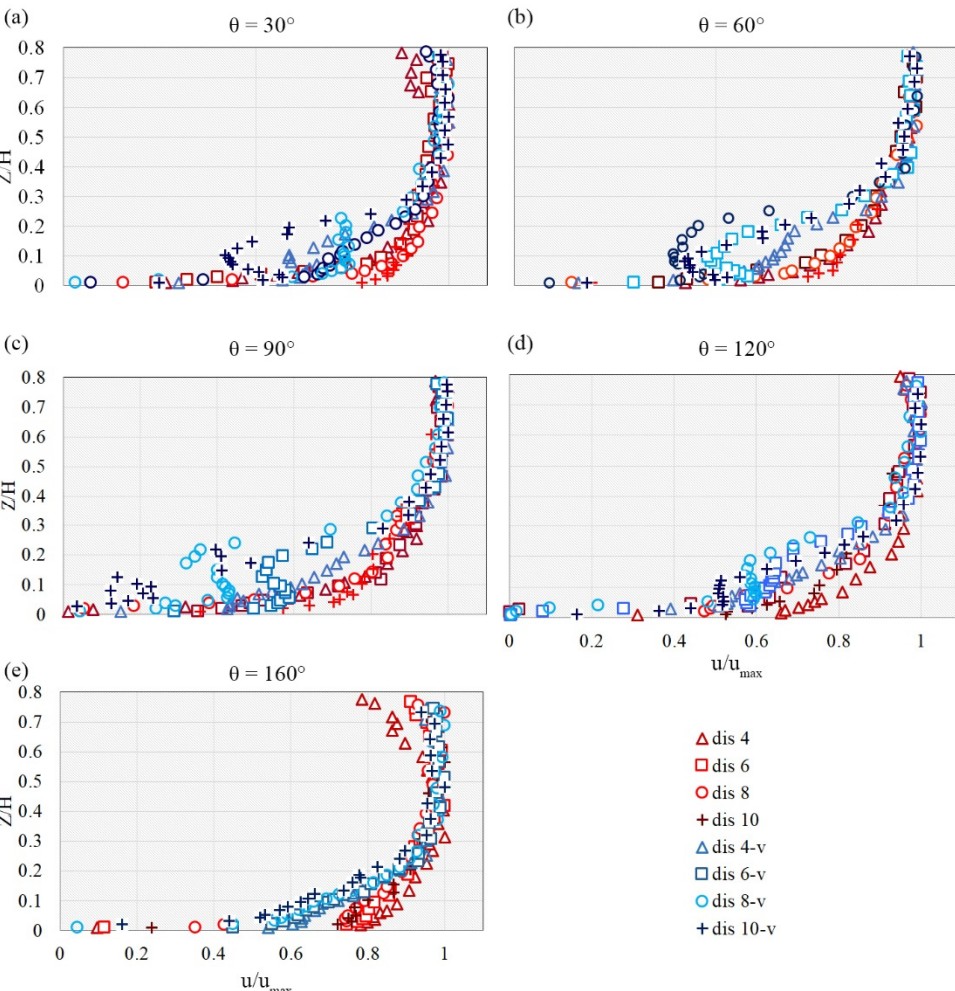

**Figure 8.** Profiles of the normalized tangential velocity *u* at different azimuthal planes, with specific distances from the abutment surface (4, 6, 8, and 10 cm) for the cases of un-vegetated bed (marked by red symbols) and vegetated bed (marked by blue symbols).

For the case with a vegetated bed, it is found that the presence of strong circulation around the abutment led to the reduction of the tangential velocity near the scoured bed. Figure 9 shows the maximum difference in tangential velocity between vegetated and un-vegetated cases. One can see that the difference in tangential velocity depended on the distance from the abutment and the azimuthal angle. The maximum difference at all angles was observed at a distance of either 8 or 10 cm from the abutment. This circulation was stronger for the case without vegetation in the bed rather than the vegetated one. It can be seen that the maximum difference occurred at 90° and the minimum difference at 160°. Further downstream, the exited flow from the scour hole joined to the main flow.

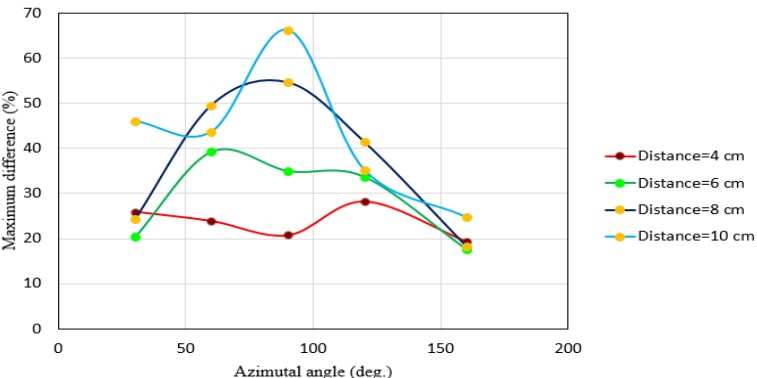

**Figure 9.** The maximum difference in tangential velocity between vegetated and un-vegetated cases of Figure 8 vs. the azimuthal angle at different distances from the abutment.

As showed in Figure 10, the radial-velocity component changed direction toward the contour line $v = 0$. This variation of the radial-velocity component at the upstream face of the abutment proved the existence of the primary vortices. Before entering the scour hole, the radial velocity was always positive; inside the hole and near the abutment base, the radial velocity had a negative and great absolute value. The flow zone with negative radial velocity created circulation due to the existence of the primary vortices. By moving from $\theta = 30°$ to $90°$, there were large negative values inside the scour hole and near the abutment. This phenomenon of reducing the negative values of $v$ with an increase in $\theta$ was noticeable from the azimuthal angle of $0°$ to $90°$ as a result of primary vortex attenuation. This decrease resulted from the vortex changes, so that at $\theta = 90°$ it covers only a small part of the scour hole, which caused a small helicoidal flow close to the abutment. The reversal in the velocity during the development of a scour hole was related to the exposure of a larger flow area. By increasing the angle from $30°$ to $90°$, the positive values were first observed in the scour hole with a distance of $r_0/3$ from the abutment and close to the edge of the scour hole. For larger angles such as $120°$ and $160°$ (moving toward the downstream region), positive radial velocities have been observed at all points. Moreover, for larger angles (moving toward the downstream region), in addition to having positive radial velocities ($v$), which is a significant factor for creating the wake vortices, the radial velocity ($v$) became stronger and had greater absolute value rather than those upstream.

The observation from the Figure 11 confirmed that the vortex intensity in regions both upstream and downstream of the abutment could be reduced by placing vegetation in the channel bed. It was noticed from the radial velocity field ($v$) that the values upstream of the abutment (the azimuthal angle ranges from $30°$ to $90°$) were negative within the scour hole ($z \leq 0$) and outside of it ($z \geq 0$) for the case with a vegetated bed, different from those of the un-vegetated bed. For the case with a vegetated bed, the radial velocity at most of the measuring points near the scour bed was less than that of the un-vegetated bed. For the case with a vegetated bed (Figure 11), the radial velocity at $120°$ also had a negative value in the upper portion of the scour hole near the abutment. At an angle of $\theta = 90°$, radial velocities had negative values for both vegetated and un-vegetated cases. The changes of signs of radial velocities showed a weak reverse flow due to the flow separation. A similar observation of the negative value of the radial velocity at a wing-wall abutment was reported by Dey and Barbhuiya [49] and Afzalimehr et al. [33]. With the increase in the radial distance from the abutment surface, the radial velocity changed to a positive and larger value. At an angle of $160°$, the radial velocity at all points was positive. The maximum positive values of the radial velocity at $160°$ for both cases (vegetated and un-vegetated) were observed in a limited region around the abutment, with a distance from 3.5 cm to 6 cm.

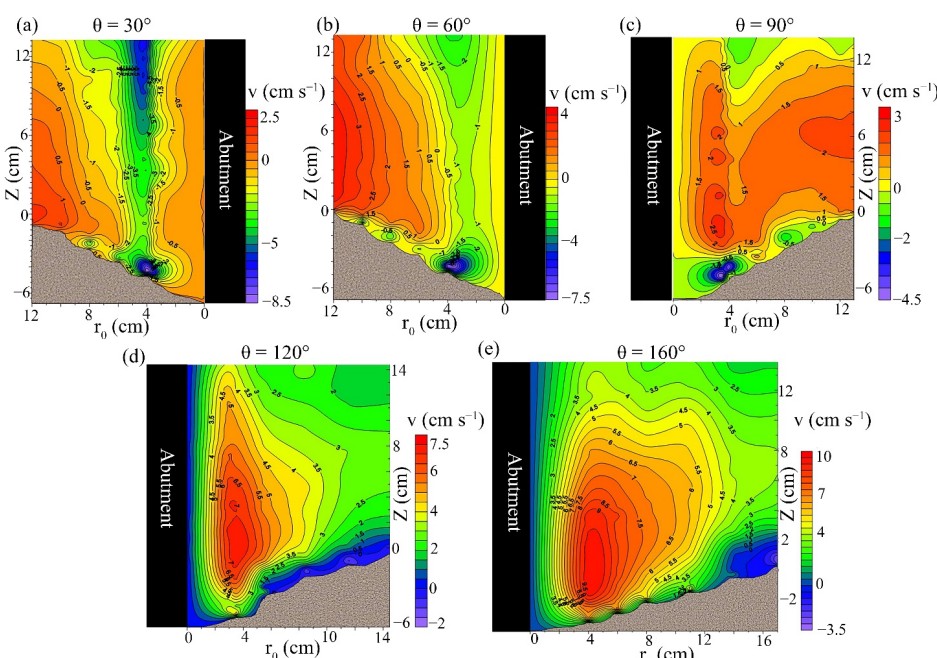

**Figure 10.** Velocity field in the radial direction (*v*, cm/s) for the un-vegetated case at different azimuthal sections of *θ*. Note that the range of the color scale is not equalized for better illustration.

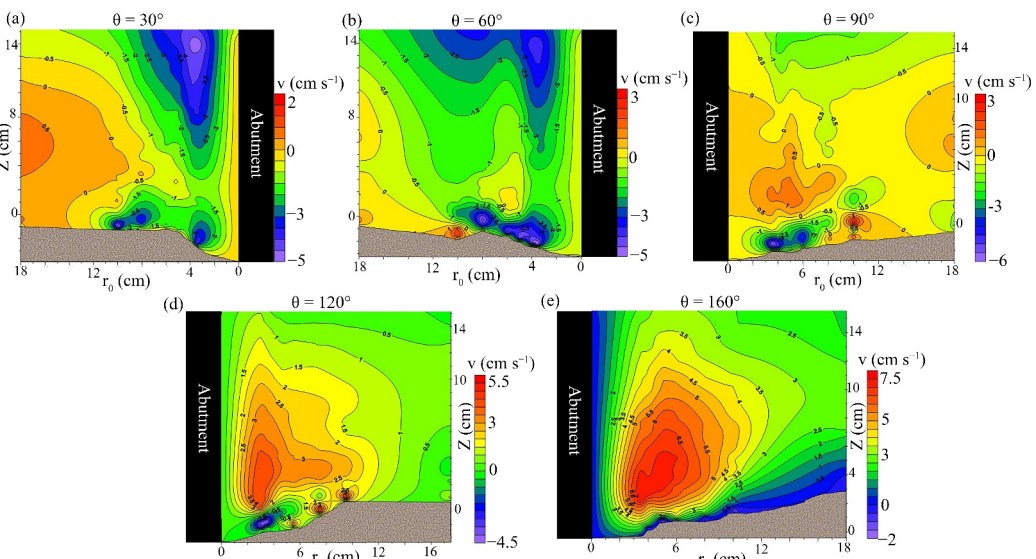

**Figure 11.** Velocity field in the radial direction (*v*, cm/s) for the case of a vegetated bed at different azimuthal sections of *θ*. Note that the range of the color scale is not equalized for better illustration.

For the case without vegetation in the bed, results presented in Figure 12 revealed that the magnitude (absolute value) of the vertical velocity (*w*) in the region upstream of the abutment increased significantly toward the scour hole bed. The results confirmed that there existed a negative pressure gradient and strong vortices in this region. The reason can be attributed to the obstruction of the flow by the abutment. On the other hand, in the downstream region (i.e., *θ* = 120° and 160°), the vertical velocity (*w*) near the scour bed became positive (with upward direction), indicating the suction process adjacent to the abutment, which is caused by separation flow. The direction of the vertical velocity (*w*) in most of the flow zone (both upstream and downstream) was downward. The maximum vertical velocity for the case without vegetation in the bed was observed at *θ* = 120°. Dey

and Barbhuiya [50] reported that the positive vertical velocities occurred at the downstream region of the abutment.

The magnitude of the negative vertical velocity was diminished upstream of the abutment and inside the scour hole (Figure 13), which is different from that for the case without vegetation in the bed, indicating that the power of the downflow leads to the primary vortices being reduced. In other words, the downflow plays an essential role during the scouring process, because the downward velocity increases and strengthens the primary vortex system, which results in a deeper scour hole [47]. For both cases of un-vegetated and vegetated beds, the reversal nature of the vertical velocity ($w$) near the scour hole bed in the upstream zone of the abutment was not distinct. However, the negative values of the vertical velocity declined with the increase in the azimuthal angle, implying a weak downward velocity, which led to the attenuation of the primary vortex toward the downstream region. For the case with a vegetated bed, at each angle, there is a spot of the maximum absolute vertical velocity ($w$) that occurred near the abutment at a depth of approximately 4 cm. Its location at the upstream region of the abutment was inside the scour hole, and with the increase of the azimuthal angle (toward downstream), the spot of the maximum absolute vertical velocity ($w$) goes to a higher depth ($z > 0$) and out of the scour hole. However, for the case with a vegetated bed, there were negative values inside the scour hole; this spot of the minimum vertical velocity is located outside of the scour hole on both upstream and downstream sides of the abutment. With the increasing azimuthal angle, the spot of the minimum vertical velocity reached close to the water's surface.

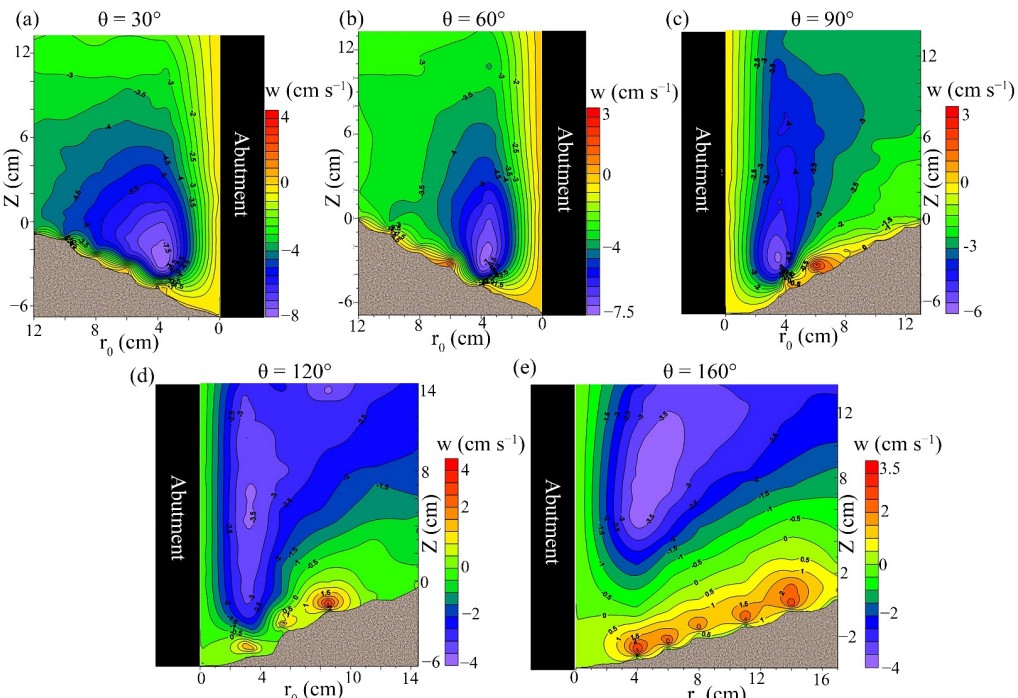

**Figure 12.** Velocity field in the vertical direction ($w$, cm/s) for the case of the un-vegetated bed at different azimuthal angles of $\theta$. Note that the range of the color scale is not equalized for better illustration.

Figure 14 displays the vertical distributions of the normalized vertical velocity component at different vertical sections. On either side of the abutment, the direction of the vertical velocity changes, showing the existence of a helicoidal flow upstream and a wake vortex near the scour hole bed. With the increase in the azimuthal angle, there will be more and more positive vertical velocities near the bed. Moreover, with the increase in the azimuthal angle, these positive values will be located further away from the scour hole bed.

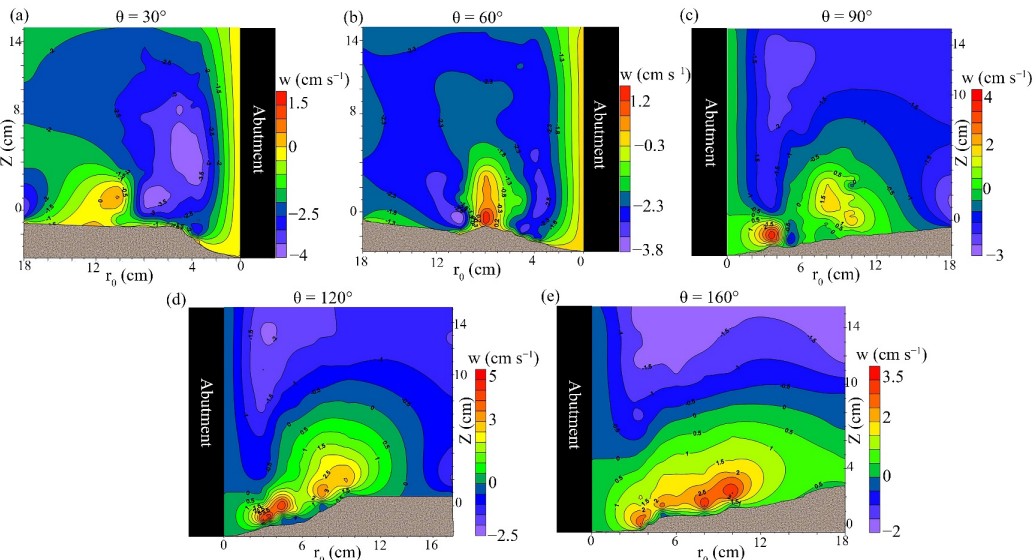

**Figure 13.** Velocity field in the vertical direction (*w*, cm/s) for the case of the vegetated bed at different azimuthal angles of *θ*. Note that the range of the color scale is not equalized for better illustration.

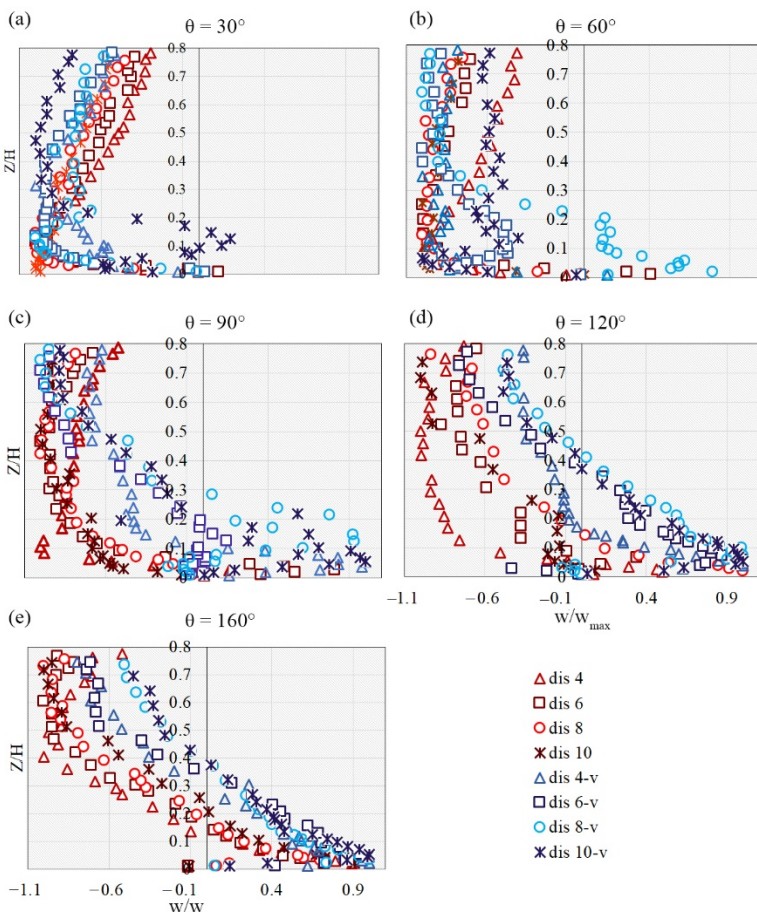

**Figure 14.** Comparison of the normalized vertical velocity (*w*) at the azimuthal planes with specific distances from the abutment surface for the un-vegetated bed (marked by red symbols) and the vegetated bed (marked by blue symbols).

For the case without vegetation in the bed, the maximum values of the absolute vertical velocity in the upstream region of the abutment occurred around the initial level

of the sand bed (the original bed level), then they moved toward the water surface. This trend changes if $\theta > 90°$. In such a way, near the scour bed, the values of $w$ become positive. With the increase in the distance from the scour hole bed, the vertical velocity around the original bed level approached zero. When the distance increased, the absolute of the negative values increased. Such a trend is observed for the case with a vegetated bed with an azimuthal angle > 90°. For the case with a vegetated bed, positive values near the scour hole and the original bed level were slightly larger than those without vegetation in the bed.

### 3.2. Reynolds Shear Stress Distribution

The vertical distribution of the normalized Reynolds shear stress ($-u'w'$) for an un-vegetated case was defined as the stress at a plane that is parallel to the flume's wall, and it is more important for us in analyzing than other Reynolds shear stresses. The normalized Reynolds shear stress distribution in Figure 15a shows the pattern of Reynolds stress at different azimuthal angles for the case without vegetation in the bed and an equal distance of 4 cm from the abutment.

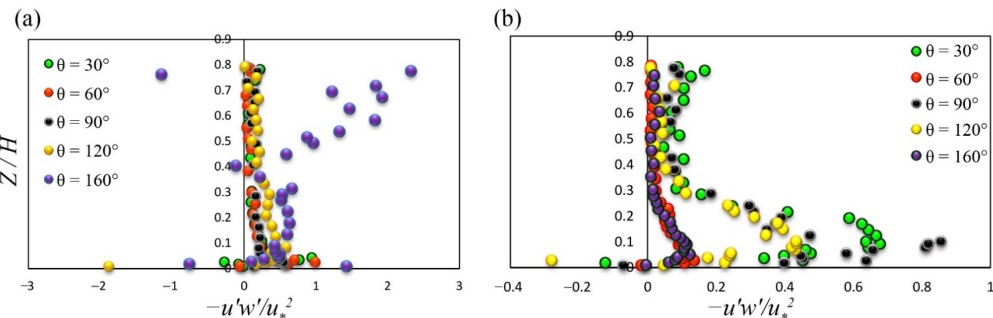

**Figure 15.** Vertical distribution of $u'w'$ at the azimuthal planes in the (**a**) un-vegetated bed and the (**b**) vegetated bed at equivalent distances from the abutment (4 cm).

Outside of the scour hole, the Reynolds stress had positive and linear distribution at all azimuthal angles (with the exception of 160°) near the original bed, which then decreased toward the water surface. However, near the scour bed it became negative, except at 90°, and was positive at other points, which could be recognized from the bulge shape in the Reynolds stress profile. In other words, the Reynolds stress had a negative value near the scour hole bed, the absolute value of the Reynolds stress declined with the increase in the (dimensionless) depth from the scour hole bed, reached zero, and then increased up to a depth around $z = 0.0255h$. Afterward, the reduction started again in the third step. This means that the maximum stress occurred at a depth around $0.0255h$. Note that this depth was not identical for different profiles. The reason for the sign-changing near the bed was a significant momentum transfer due to convective acceleration.

It should be noted that the negative Reynolds shear stress indicated the reverse flow, and the point where the sign changed from the positive to the negative indicated the separation point. The maximum value of the normalized Reynolds shear stress ($-u'w'$) near the scour bed was $1.43u^{*2}$, located at $\theta = 160°$ in the radial distance from the abutment and with $z = 0.2$ cm. The high values of Reynolds shear stress could be attributed to the high-pressure gradient and the effort of the flow for separation. Both stress and the high-pressure gradient above the scour bed were partially responsible for developing the primary vortex and the scour hole. In the downstream region (at $\theta = 160°$), the Reynolds stresses did not follow a continuous trend. The shear stress increased dramatically at this angle and reached its maximum value near the water surface at a depth of $z = 0.775h$.

Dey and Barbhuiya [50] pointed out that Reynolds stress had uneven distribution and an unknown pattern in the region downstream of the abutment. They attributed this phenomenon to the flow separation and vortex shedding in this region. In the present study, this point was observed at $\theta = 160°$ (Figure 15a), and there was no specified collocation

at this azimuthal angle. For the case with a vegetated bed (Figure 15b), a special pattern has been observed at most of the azimuthal angles (except $\theta$ = 90°). One can see from Figure 15b that, inside the scour hole, the vertical distribution of the normalized Reynolds shear stress has a bulges shape.

For the case with a vegetated bed, the vertical distribution of the normalized Reynolds shear stress above the scour hole had an approximately linear trend and decreased toward the water surface, although at some angles it increased near the water surface. For the case with a vegetated bed, the magnitude of $-u'w'$ near the scour bed and above it decreased (except in $\theta$ = 90°) with the increase in the azimuthal angle, indicating the decline of the primary vortex and pressure gradient around the abutment. In general, by comparing the Reynolds shear stress ($-u'w'$) for the case without vegetation in the bed to that for the case with a vegetated bed, one could say that the maximum stress near the bed occurred at $\theta$ = 160° for the case without vegetation in the bed (which also had a significant value), while the maximum stress was observed at an azimuthal angle of 90° for the case with the vegetated bed.

The distributions of Reynolds shear stresses ($-u'w'$) for 40 measuring positions were presented in Figure 16. Toward the scour hole, the velocity decreases with the increase in the flow depth, leading to $dp/d\theta > 0$. Therefore, the trend of the pressure gradient and shear stress near the scour hole bed is positive, presented by a convex distribution curve. A convex distribution indicates a decelerating flow due to the generation of turbulence. For the case without vegetation in bed, the convex distribution of Reynolds stress was observed at all angles from 0° to 120°, except at a distance of 10 cm from the abutment surface. At the angle of 160°, it is observed that at certain distances away from the abutment ($d$ = 8 and 10 cm), the distributions of Reynolds shear stresses ($-u'w'$) are similar to the convex shape. While for the case with the vegetated bed, the distributions of Reynolds shear stresses ($-u'w'$) have a convex shape at all azimuthal angles and follow a certain trend. For the case with the vegetated bed, a convex distribution of Reynolds stress at 160° has an increasing value compared to those at other angles. The comparison of Reynolds shear stress for the vegetated case to that for the un-vegetated case indicates that vegetation in the channel bed dramatically reduces the Reynolds shear stress component. The Reynolds shear stresses ($-u'w'$) for the case with the vegetated bed, at all angles except at ($\theta$ = 160°, $r_0$ = 4 cm), are less than those for the case with the un-vegetated bed. Moreover, the Reynolds shear stress ($--u'w'$) shows a decreasing trend in the vertical direction toward the water surface from a depth of $Z/H > 0.05$ at the upstream surface of the abutment and from a depth of $Z/H > 0.14$ at the downstream surface of the abutment. In the scour hole, however, the Reynolds shear stress ($-u'w'$) displays an increasing trend in the vertical direction from the scour bed toward a depth of $Z/H$, as mentioned above. Siniscalchi et al. [51] stated that a negative Reynolds shear stress ($-u'w'$) indicates an upward vertical momentum transport with negative velocity gradients. The magnitude of negative Reynolds shear stress ($-u'w'$) can be seen in the downstream side of the abutment ($\theta > 90°$), and their values increase with the increase in the azimuthal angle ($\theta$). Such Reynolds stress $-u'w'$ near the scour bed confirms the development of the wake zones downstream of the abutment, which occurs at 120° for the case with the vegetated bed and 160° for the case without vegetation in the bed. These wake zones lead to the generation of strong turbulence.

### 3.3. Turbulence Intensity

The vertical distributions of the normalized tangential, radial, and vertical turbulent intensity components are shown in Figure 17.

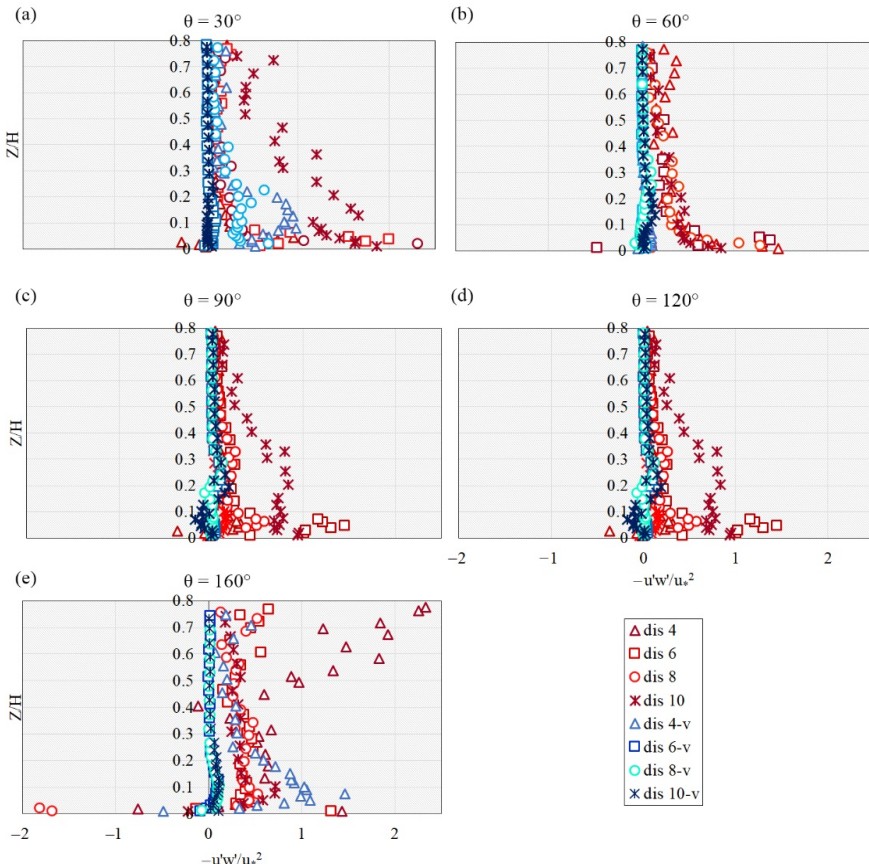

**Figure 16.** Comparison of the Reynolds shear stress distribution at the azimuthal planes with specific distances from the abutment surface (4, 6, 8, and 10 cm) for the case without vegetation in the bed (marked by red symbols) and with vegetation in the bed (marked by blue symbols).

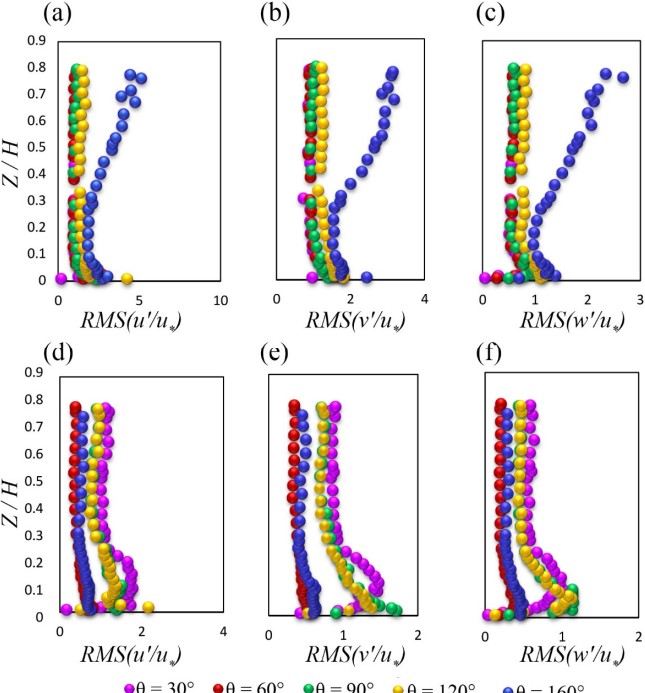

**Figure 17.** Turbulence intensity distributions in 3D at different azimuthal sections in the (**a**–**c**) channel bed without vegetation and (**d**–**f**) the channel bed with vegetation, at distances of 4 cm from the abutment.

The turbulence intensity values in both tangential and radial directions were greater than those in the vertical direction. For the case with vegetation in the bed, turbulence intensities were lower than those for the case without vegetation in the bed, with the exception of inside the scour hole at the azimuthal angles of $\theta = 30°$ and $\theta = 90°$. The turbulence intensity values decreased with the increase in the distance from the scour bed and changed to a linear trend (except at $\theta = 160°$ for the case without vegetation in the bed). The maximum value of turbulence intensity was observed at the angle of $160°$ for the case without vegetation in the bed.

*3.4. Reynolds Stress Anisotropy*

The Reynolds averaged Navier–Stokes equation is used to define the normalized anisotropy tensor $b_{ij}$. The difference between the ratio of Reynolds stress tensor terms to turbulence kinetic energy (TKE) and its isotropic equivalent quantity gives the Reynolds stress anisotropy tensor $b_{ij}$, which provides an estimate of the degree of departure from the idealized isotropic turbulence [20,52]. The $b_{ij}$ is given by:

$$b_{ij} = \frac{\overline{\acute{u}_i \acute{u}_j}}{2k} - \frac{1}{3}\delta_{ij} \tag{2}$$

where $\acute{u}_i$ is the instantaneous velocity fluctuation in the direction $i$, $\delta_{ij}$ is the Kroecker delta function, which is $\delta_{ij}(i \neq j) = 0$ and $\delta_{ij}(i = j) = 1$, $k$ is the turbulent kinetic energy of the flow defined as $k$ = average TKE = $0.5(\overline{\acute{u}\acute{u}} + \overline{\acute{v}\acute{v}} + \overline{\acute{w}\acute{w}})$, and $i, j$ = 1, 2, 3 are the spatial components. The normalized anisotropy tensor $b_{ij}$ has a zero trace as a consequence of its formulation, which two independent invariants can represent. These invariants are defined as follows:

$$\begin{aligned} I_2 &= -b_{ij}b_{ji}/2 \\ I_3 &= -b_{ij}b_{jk}b_{ki}/3 \end{aligned} \tag{3}$$

The anisotropy invariant map (AIM) of Lumley and Newman [20], also called the "Lumley triangle", uses this second $I_2$ and third principal $I_3$ components of turbulence anisotropy to create the coordinate system ($I_3$, $I_2$). Another method that can construct a nonlinear anisotropy invariant map (AIM) is based on the invariants $I_2$ and $I_3$ called the "Turbulence triangle", which in general is also termed the Lumley triangle and uses the coordinate system ($\xi$, $\eta$), where:

$$\begin{aligned} \xi &= (I_3/2)^{1/3} \\ \eta &= (-I_2/3)^{1/2} \end{aligned} \tag{4}$$

When plotting $\xi$ vs. $\eta$, the domain of both invariants confined by three lines is reduced to the interior of a triangle [52]. The triangle boundaries define several characteristics of turbulence states classified based on the shape of the eddies. In the AIM, it is observed that three vertices and edges of the triangle correspond to isotropic turbulence (denoted by 3D) where the three normal stresses are equal, one-component isotropic turbulence (denoted by 1D), and two-component isotropic turbulence (denoted by 2D), respectively (Figure 18). The limitation of turbulent structures to two distinct types transfers turbulence from the 3D to 2D and/or 1D: (1) the left side of the triangle corresponds to pancake-shaped turbulence, where the fluctuations of turbulence exist along with two directions with equal magnitude. These two equal components (which show with $\sigma_i$ as Reynolds normal stress) have a considerably higher amplitude than the small one ($\sigma_1 = \sigma_2 > \sigma_3$), and this state is known by 2D turbulence; (2) the right side of the triangle corresponds to a rod-like or cigar-shaped turbulence where the turbulent fluctuations only exist along one direction (1D), in which one principal component is larger than the other two equal components ($\sigma_1 = \sigma_2 < \sigma_3$). The upper boundary curve of the triangle, defined by ($\eta^2 = (1/27 + 2\xi^3)$), is used to represent the isotropy of the two-component turbulence. Briefly, as it has been said, the turbulence triangle consists of bottom, right, and left vertices (3D, 1D, and 2D, respectively) bounded by three boundaries (two linear boundaries known as axisymmetric

turbulence and one nonlinear boundary). The region within this triangle and far from the specified limit indicates the general tridimensional turbulence condition. Any data points that lie within this limitation of the turbulence triangle refer to a specific turbulence state.

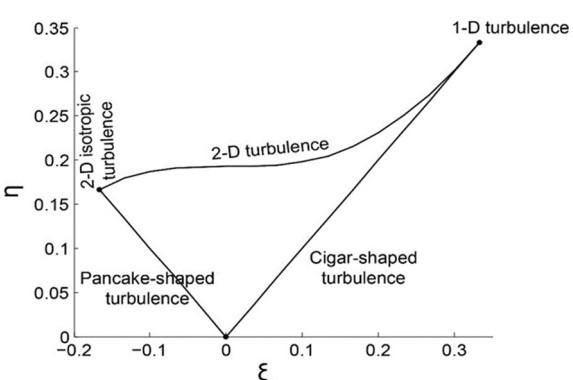

**Figure 18.** A schematic shape of a Lumley triangle in the $\eta$ (2nd invariant) and $\xi$ (3rd invariant).

For deeper insight into the anisotropy level, an invariant function ($F$) is expressed as the following [53]:

$$F = 1 + 9I_2 + 27I_3$$
$$\to If : \begin{cases} F = 0 \to \text{Two dimensional (top side of the triangle)} \\ F = 1 \to \text{ Isotropic turbulence (bottom vertex of the triangle)} \end{cases} \tag{5}$$

The invariants $\xi$ and $\eta$ have been plotted on the Lumley triangle for both cases with vegetation and without vegetation in channel the bed (Figure 19). The evolution of the invariants $\eta$ and $\xi$ across the z-direction has been analyzed at different azimuthal angles (marked by different symbols) and radial distances from the abutment (4, 6 (or) 8, and 10 cm, highlighted by different colors). By comparing Figure 19a to Figure 19b, it is realized that the invariants for the case with vegetation in the bed showed the tendency to be closer to the origin than that of the case without vegetation. This result indicated that it is slightly closer to isotropic for the vegetated bed than that without vegetation, because the placement of the vegetation on the bed decreased $\tau_{uw}$ around the abutment, especially inside the scour hole. One can also see from Figure 19 that there was both pancake-shaped and cigar-shaped turbulence, based on the data measured in the laboratory and according to the spread of points in the respective turbulence triangles. As can be seen in Figure 19, most of the anisotropy invariants for both cases (with and without vegetation in the bed) have been gathered between the 2D and the cigar-shaped limit. In the triangle for the un-vegetated case, it is observed that the level of anisotropy increases with an increase in the distance from the abutment.

The majority of the values of $\xi$ were less than zero, and a trend toward pancake-shaped turbulence has been observed (Figure 19) (even by increasing the radial distance). At the azimuthal angle of $\theta = 90°$, the interaction region appeared, which showed a tendency toward the cigar-shaped boundary. In the region downstream of the abutment, there was no trend at the angles of 120° and 160° (except at the distance of 4 cm) caused by the separation of flow. Note that the turbulence anisotropy at the angle of 160° and close to the abutment (with a distance of 4 cm from the abutment) was in the opposite direction of the trend for the angle of 30° (with the same distance). This state at this angle ($\theta = 160°$) is also called rod-like or cigar-shaped turbulence.

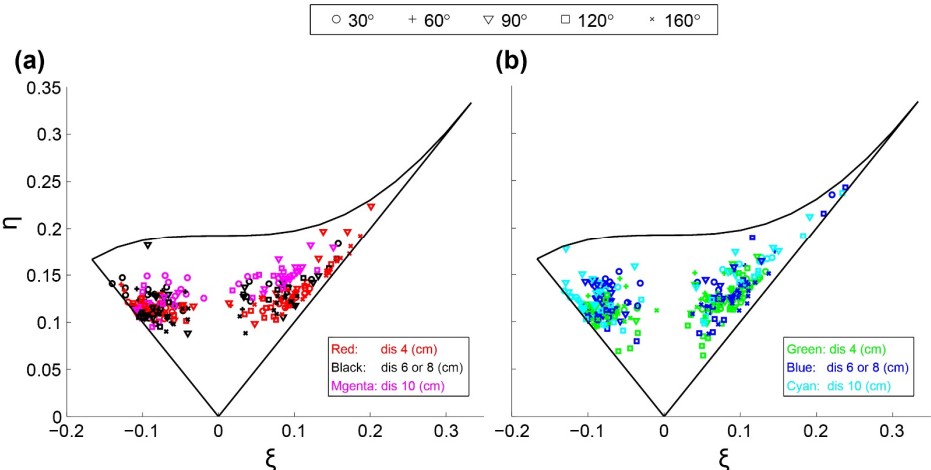

**Figure 19.** Turbulence triangle plotted in the $\xi - \eta$ plane at specified azimuthal sections in the case (**a**) without vegetation in the bed and (**b**) with vegetation in the bed, at radial distances from the abutment.

For the vegetated channel bed (Figure 19b) in the upstream region of the abutment (i.e., $\theta = 30°$ and $60°$), an opposite trend has been noticed compared to the case without vegetation in bed. For the majority of the values of $\xi$ that were greater than zero and at an azimuthal angle of $30°$, there was a tendency to be cigar-shaped. At an azimuthal angle of $90°$, similar to the case of an un-vegetated bed, the pattern of the turbulence anisotropy changed, data points lay on the left line curve ($\xi < 0$). In the region downstream of the abutment, a similar trend toward that for the un-vegetated bed occurred. Note that for the angle of $160°$ and with a distance 4 cm from the abutment, the data points lay close to the left line curve ($\xi < 0$) for the point near the scour bed.

As showed in Figure 20, the turbulence anisotropy across the vertical direction has been evaluated at a constant distance from the abutment (4 cm) at different azimuthal planes. This figure uses two different symbols (circles and squares) for un-vegetated and vegetated beds, respectively. The movement of the anisotropy vs. depth, fitted with two curves of black and gray color with arrowheads, is illustrated.

For the case with an un-vegetated bed, with the increase in vertical distance, the turbulence anisotropy invariants change from pancake-shaped to cigar-shaped in the front face of the abutment. Namely, upstream of the abutment near the scour bed, the anisotropy tends to be pancake-shaped, and its values increase with the increase in the azimuthal angle. Eventually, near the water surface, the turbulence anisotropy reaches the right boundary at which one of the components of TKE is larger than the other two. At the azimuthal angle of $90°$, the anisotropic turbulence state and kinetic energy act inversely. Thus, there is a trend toward the cigar-shaped boundary with a high anisotropic level near the scour bed. At the downstream side of the abutment with an azimuth angle of $160°$, the turbulence anisotropy is more uniform, which shows a tendency toward the cigar-shaped boundary. Moreover, the level of anisotropy increases with the depth increase and follows a path almost parallel to the right boundary. Therefore, as presented in Figure 20, it is clear that the data points of the surface zone at all angles for the case without vegetation in the bed approach the cigar-shaped limit.

The degree of turbulence anisotropy for the case with vegetation in the bed, close to the abutment (4 cm) at the upstream region and inside the scour hole, is higher than that for the case without vegetation in the bed. In addition, unlike the turbulence anisotropy for the case without vegetation in the bed, the turbulence anisotropy for the case with vegetation in the bed starts from the cigar shape, and then the turbulence anisotropy moves to the opposite direction as the vertical distance increases. The tendency to the cigar-type structure near the scour bed in the region upstream of the abutment indicates the presence of a dominant direction of velocity fluctuations. At an azimuthal angle of $\theta = 90°$, similar

to a trend for the un-vegetated case, an opposite trend toward the upstream region of the abutment is observed near the scour bed, which shows a tendency toward the pancake anisotropy limit. In the region downstream of the abutment ($\theta$ = 120°, 160°), different from the results for the case with the un-vegetated bed, no definite relation between the spatial distribution of the turbulence and the vertical position is observed either near the scour hole or at the water surface. Moreover, the turbulence anisotropy pattern rotates from the pancake shape to the cigar shape from the bottom towards the surface.

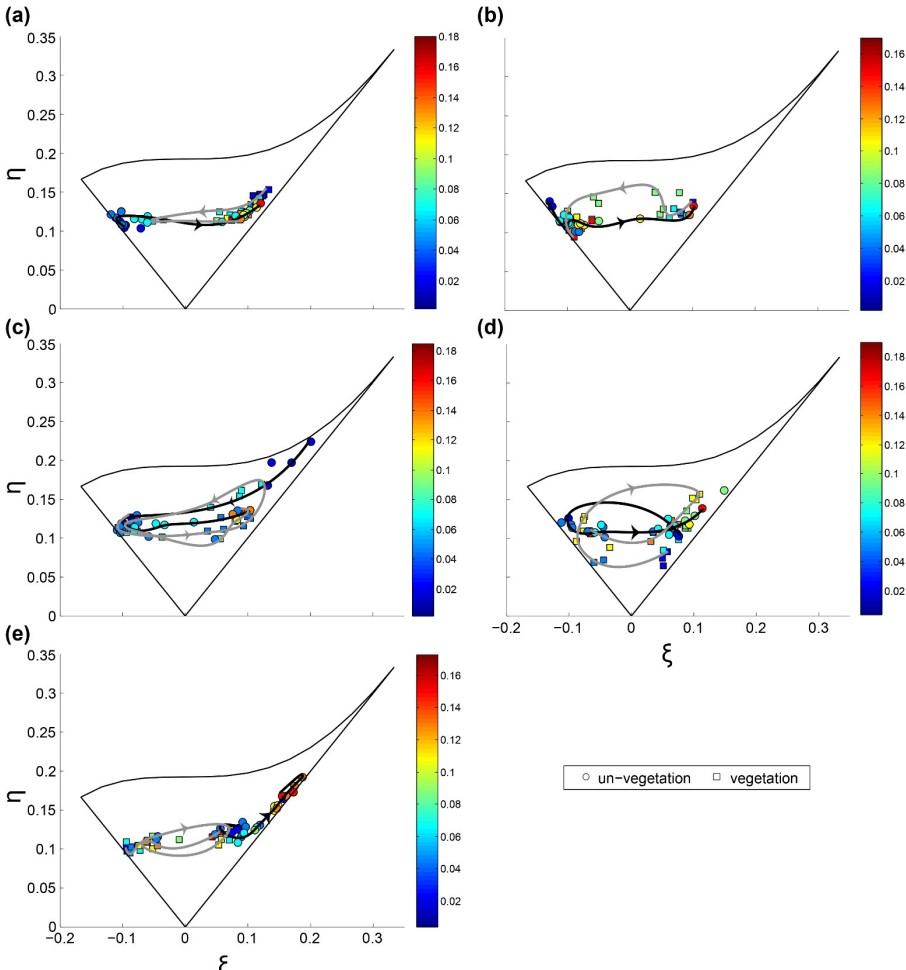

**Figure 20.** Maps of the anisotropic invariant for the cases of the un-vegetated bed (labeled by circular shapes) and vegetated bed (labeled by rectangular shapes) at the azimuthal sections of $\theta$ = (**a**) 30°, (**b**) 60°, (**c**) 90°, (**d**) 120°, (**e**) 160° with a constant distance from the abutment (4 cm). The depth of the measurement is color-coded in meters. The distribution of the turbulence anisotropy for both cases of un-vegetated and vegetated bed throughout the water column is denoted by black and gray curves, respectively. The direction from the bottom to the water surface is indicated by the arrowheads.

On the other hand, the degree of the turbulence anisotropy downstream of the abutment near the scour hole has its lowest value. For all angles (except for $\theta$ = 30°), it has been observed that if data points near the scour bed approach one of the anisotropy types, it tends to the opposite region near the water surface. For instance, near the scour hole at $\theta$ = 160°, data points are confined to the left side of the Lumley, whereas the approach to the right side of the Lumley is near the surface zone. The highest degree of anisotropy for the un-vegetated bed usually occurs at the points near the scour hole and/or at the points near the scour hole or the water surface. This result was also reported by Mera et al. [24]. Note that the highest degrees of anisotropy occur near the water surface in the region downstream of the abutment. While for the case with vegetation in the bed, at $\theta$ =

30°, the highest degree of anisotropy is observed near the scour bed. With the increases in the angle, the highest degree of anisotropy is located out of the scour hole. In the region downstream of the abutment, the lowest degree of anisotropy occurs near the scour bed.

To preciously assess the turbulence anisotropy changes at specific angles with a constant distance from the abutment (4 cm) along the tangential direction, a parameter named anisotropic invariant function (*F*) was defined here. Figure 21 shows the anisotropic invariant function (*F*) for the flow depth (*Z/H*). At the upstream face of the abutment, the value of the invariant function (F) near the scour hole for the case of the un-vegetated bed is more than that for the case with vegetation in the bed. At all angles, inside the scour hole and near the scour bed, the value of F is close to 0.7 for both vegetated and un-vegetated cases. However, at an angle of θ = 90° and with a very close distance to the scour bed, the value of F for an un-vegetated bed is 0.08, which corresponds to two-dimensional isotropy. Downstream of the abutment, the opposite condition happens, and the value of F for the case of the vegetated bed near the scour hole is greater than that for the case of the un-vegetated bed. Moreover, at an angle of θ = 120° inside the scour hole and near the scour bed, the value of F indicates three-dimensional isotropy.

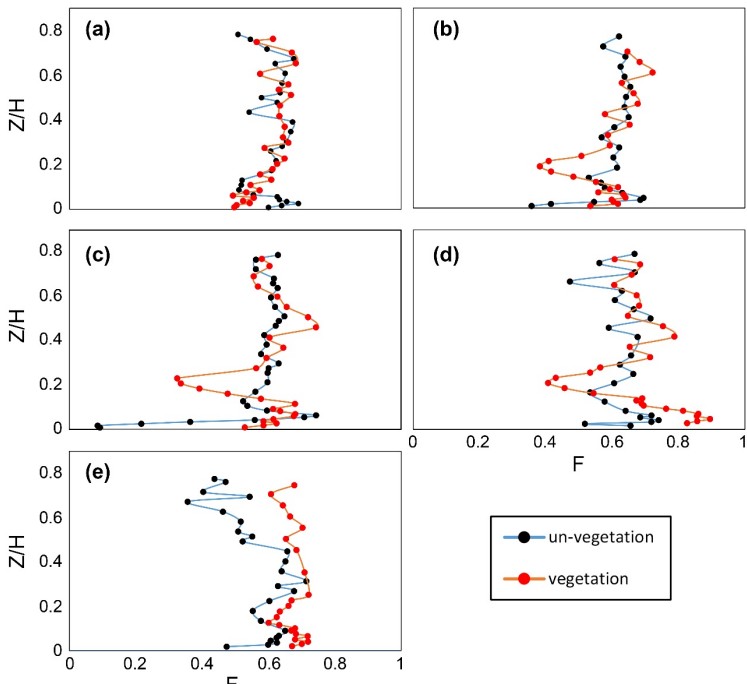

**Figure 21.** Evolution of the invariant function (*F*) at the azimuthal sections of θ = (**a**) 30°, (**b**) 60°, (**c**) 90°, (**d**) 120°, and (**e**) 160° for the cases of the un-vegetated bed and the vegetated bed at an equal distance of 4 cm from the abutment.

As the vertical distance approaches very close to the scour bed at all azimuthal angles, the turbulence anisotropy tends to return to the isotropic limit. Then, the turbulence anisotropy moves away from the isotropic limit with a gradual increase in vertical distance, and the invariant function (*F*) decreases. The changes of the invariant function (*F*) were observed between 0.5 and 0.7, and the invariant function (*F*) increased with the angle (except for 160°) for the case of the vegetated bed.

## 4. Conclusions

The following conclusions can be drawn from this study:

(1) Compared to the case of an un-vegetated bed, with vegetation around the abutment, the maximum scour depth occurred at the abutment tip instead of the upstream nose of the abutment. The presence of vegetation in the channel decreases the maximum

values of the 3D velocity components (except for the positive values of $w$ downstream and near the scour bed) and reduces the turbulence intensities inside the scour hole. For this experimental study, the presence of vegetation in the channel bed caused a reduction of 34.8% of the scour depth. The time required for achieving the equilibrium condition also decreased. Interestingly, within the first 120 min of the experiments, the scour depth for the case with vegetation in the bed was slightly higher than that for the case without vegetation.

(2) The circulation and vortices have been weakened due to the presence of vegetation in the channel bed. In other words, the primary vortex for the case of an un-vegetated bed was stronger than that for the vegetated bed. The turbulence kinetic energy inside the scour hole is reduced.

(3) The tangential velocity component was always positive and decreased rapidly near the scour bed, especially around the upstream face of the abutment. Additionally, the deceleration process of flow has proceeded slowly with vegetation in the channel bed. The tangential velocity increased with the increase in the azimuthal angle for both vegetated and un-vegetated cases, but the rate of deceleration at $\theta = 90°$ for the vegetated case was greater than those of $\theta = 30°, 60°$. The maximum tangential velocity difference between two cases was observed at an angle of 90°. With vegetation's presence, the tangential velocity profile (at all angles except the 160°) near the bed had changed to an "S" shape, which showed two turning points. Near the vegetated bed, the tangential velocity gradient with radial distance from the abutment became negative, especially between 5 cm to 12 cm (except at 160°).

(4) The contours of radial velocity (v) upstream of the abutment (30° to 90°) had negative values for the case with vegetation in the bed within the scour hole ($z \leq 0$) and above it ($z \geq 0$), different from that for the case of un-vegetated bed. Downstream of the abutment, the sign change in radial velocity was observed for the case of the un-vegetated bed only at 90° and for the case of the vegetated bed at $\theta = 90°, 120°$. This indicated a week flow reversal as a result of backflow. For the vegetated bed, variation in radial velocity inside the scour hole was less than that for the un-vegetated one (except 90° near the scour bed), indicating reduction in the primary vortex by vegetation.

(5) For both vegetated and un-vegetated cases, the magnitude of the vertical velocity ($w$) toward the scour bed increased significantly, resulting in the powerful downflow around the abutment. For the case with vegetation in the bed inside the scour hole upstream of the abutment, the magnitude of the negative vertical velocity is diminished in this region compared to that for the un-vegetated bed, indicating that the power of downflow leads to a reduction of the primary vortices. The extent of the positive vertical velocities for the case with vegetation in the bed was increased and covered broader if the azimuthal angle was more than 90°, compared to the un-vegetated case. The maximum positive vertical velocities were observed at 160° for the un-vegetated bed and 120° for vegetation. It is found that vegetation could not significantly reduce the effect of upward flow in the downstream region adjacent to the abutment due to the suction that takes place.

(6) Although radial and vertical velocities played an utmost important role in the flow field, the effect of tangential velocity was the highest, implying that the tangential velocity ($u$) contributed more to the turbulence intensities, Reynolds shear stress, and consequently the development of the scour holes.

(7) The presence of vegetation in the bed dramatically reduces the Reynolds shear stress component, except at the distance of 4 cm and 160° near the scour bed. For the case of the un-vegetated bed, the convex distribution of Reynolds shear stress ($-u'w'$) was observed at all angles from 0° to 120°, except at a distance of 10 cm. With vegetation, Reynolds shear stress profiles were convex at all azimuthal angles and had a certain trend. The magnitude of negative Reynolds shear stress ($-u'w'$) confirms the development of wake zones downstream of the abutment, which occur at 120° for the

vegetated bed and 160° for the case of the un-vegetated bed. These wake zones lead to the generation of strong turbulence. At a constant distance of 4 cm from the abutment, inside the scour hole, the vertical distribution of Reynolds shear stress and turbulence intensity possessed a convex shape. Near the scour bed, with the increase in the azimuthal angle (except 90°), both Reynolds shear stress and turbulence intensity decreased for the case of the vegetated bed. For the case of an un-vegetated bed, the maximum stress occurred near the bed at an angle of $\theta = 160°$.

(8) By placing vegetation in the bed, for the vegetated bed, the flow tends to be more isotropic than that for the un-vegetated case. For the un-vegetated bed, the anisotropy level increases with the radial distance from the abutment due to the turbulence triangle. Moreover, at the upstream face of the abutment, most of the data points in the Lumley triangle tend to be pancake-shaped turbulence. However, for the case of a vegetated bed, there was a tendency to be cigar-shaped. Downstream of the abutment, due to flow separation, there was no definite trend. For the un-vegetated bed, upstream of the abutment, the anisotropy at a close distance of 4 cm from the abutment changes from pancake-shaped to cigar-shaped with the increase in the vertical distance (from the scour bed). For the case of a vegetated bed, the anisotropy starts from the cigar shape then changes to the opposite direction. Inside the scour hole in this zone, the degree of anisotropy for the case of the vegetated bed is larger than that for the un-vegetated bed. Downstream of the abutment, for the case of the un-vegetated bed, the anisotropy of turbulence is more uniform at $\theta = 160°$ and presents a tendency toward the cigar-shaped boundary. The highest degree of anisotropy for the case of an un-vegetated bed occurs near the scour hole or near the water surface. The vegetated bed was observed near the scour bed at $\theta = 30°$, but it was located outside of the scour hole by increasing the angle.

(9) For the case of an un-vegetated bed, inside the hole upstream of the abutment, the invariant function (F) value was greater than that for the vegetated bed. Downstream of the abutment, however, it was the opposite. Moreover, the F value for the case of the un-vegetated bed near the scour hole at $\theta = 90°$ corresponds to two-dimensional isotropy. For the vegetated bed, however, inside the hole at $\theta = 120°$, F leads to three-dimensional isotropy, with the highest amount of all the azimuthal angles for both cases.

**Author Contributions:** Conceptualization, S.F.N., H.A., J.S., B.K. and S.H.N.; methodology, S.F.N., H.A. and B.K.; validation, S.F.N., H.A., J.S., B.K. and S.H.N.; formal analysis, S.F.N., H.A. and S.H.N.; investigation, S.F.N., J.S. and B.K.; resources, S.F.N., H.A., J.S., B.K.; data curation, S.F.N. and S.H.N.; writing—original draft preparation, S.F.N. and S.H.N.; writing—review and editing, H.A., J.S. and B.K.; visualization, S.F.N.; supervision, H.A., J.S. and B.K.; funding acquisition, H.A. and J.S. All authors have read and agreed to the published version of the manuscript.

**Funding:** This research received no external funding.

**Institutional Review Board Statement:** The study does not involve the use of humans or animals.

**Informed Consent Statement:** Not applicable.

**Data Availability Statement:** The data presented in this study are available on request from the first author.

**Conflicts of Interest:** The authors declare no conflict of interest.

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
