# Peer review of "Investigation of the Effect of Vegetation on Flow Structures and Turbulence Anisotropy around Semi-Elliptical Abutment"

_water, doi:10.3390/w13213108_

Round 1
Reviewer 1 Report
The paper has a total of 29 pages and it follows this structure:
- Introduction
Where the authors refer the importance of their study namely on the flow structures near abutments and vegetated beds.
- Materials and Methods
Where the experimental setup is presented. Authors carried out these experiments in a laboratory flume with an abutment and with and without vegetation. A downlooking ADV probe was used to measure instantaneous velocities.
- Results and discussions
Where the authors present their results including the velocity field around the abutment, the Reynolds shear stress distribution, the turbulence intensity, Reynolds stress anisotropy
- Conclusions
FINAL OF THE REVIEW
Comments on the paper:
Line 15: It was not only the flow structure but also the scour profile.
Line 30: r0 was not presented before.
Line 38: Is there a need for including 12 references for the importance of the scour processes for designing bridges?
Line 58: When the vortices are explained, it would be good to have a figure. The book of Melville has this flow structure well represented.
Line 79: Correct “The purpose fo”
Line 112 : In the introduction (and along the paper) it is important to distinguish between surface and easily submerged vegetation and vegetation like trees or shrubs.
Line 127: It is not clear the main innovative issue as the authors previously have referred the studies of Afzalimehr et al. [40], Keshavarz et al. [41, Afzalimehr et al. [42].
Line 176: Correct seiment
Please include the Froude number of the experiment
Line 179: I guess the experiment was subcritical. Please replace the word critical flow depth.
Line 179: If the slope and the discharge were not changed how did the authors change the water depth? My guess is that there was a tailgate downstream (that was moved) but this was not presented in the paper.
It is not clear the density of the vegetation.
It was not clear if the authors added seeding to the water
Figure 3 would beneficiate with the inclusion of the abutment
This sentence needs to be rewritten: “(the minimum particle size that the ADV manufacturer Sontek recommends)”
Sontek is not the manufacturer of Vectrino ADV probe.
Line 282: These temporal changes and the processes could be presented in figure 4.
In my opinion, the paper goes too deep in the results. As this is case specific, I think it is more important to have the big picture on the phenomena rather than have the all values and results. For instance, is it important to show in figure 9 the maximum differences?
Line 777: Correct Fro
In section 3.4, authors frequently refer to pancake or cigar shape. In my opinion, authors should stress more the effects on the 1d/2d turbulence and link it with the vegetation.
Reviewer 2 Report
In the proposed work, the effect of vegetation on flow structure around bridge abutment has been investigated. The work is of experimental nature and a semi-elliptical abutment model has been used. Equivalent experimental conditions have been setup by authors for comparing the effect of vegetated channel bed on the local scour around abutments to that of without a vegetation cover.
The topic is of interest and the proposed objective is ambitious.
The Introduction section is clear and rather complete. The problem at hand is well stated and the motiviation for this study is clear.
Some papers are reported as a simple list (e.g. [1-12], [19-24], [34-39]), try to reduce.
I suggest to include the recent paper:
Antonino D’Ippolito, Francesco Calomino, Giancarlo Alfonsi & Agostino Lauria (2021) Drag coefficient of in-line emergent vegetation in open channel flow, International Journal of River Basin Management, DOI: 10.1080/15715124.2021.1961796
Page 2 line 28 replace “Kwan” with “Kw An”
Also the section Materials and Methods seems to be clear and rather complete.
At page 6 (figure 3) authors shows the topography of the final scouring hole around the abutment without vegetation and with vegetation. Add details about the methodology used to obtan topography.
The Results and Discussion sections seems to be clear, figures are easy to interpret showing essential data.
The Conclusions section seem to be supported by approprite evidence.
